# Protective Effects of Bee Pollen on Multiple Propionic Acid-Induced Biochemical Autistic Features in a Rat Model

**DOI:** 10.3390/metabo12070571

**Published:** 2022-06-21

**Authors:** Hanan A. Alfawaz, Afaf El-Ansary, Laila Al-Ayadhi, Ramesa Shafi Bhat, Wail M. Hassan

**Affiliations:** 1Department of Food Science and Nutrition, College of Food and Agriculture Sciences, King Saud University, Riyadh 11495, Saudi Arabia; halfawaz@ksu.edu.sa; 2Central Research Laboratory, Female Center for Medical Studies and Scientific Section, King Saud University, Riyadh 11495, Saudi Arabia; 3Department of Physiology, Faculty of Medicine, King Saud University, Riyadh 11461, Saudi Arabia; ayadh2@gmail.com; 4Biochemistry Department, College of Sciences, King Saud University, Riyadh 11495, Saudi Arabia; rbhat@ksu.edu.sa; 5Department of Biomedical Sciences, School of Medicine, University of Missouri Kansas City, Kansas City, MO 64108, USA; hassanwm@umkc.edu

**Keywords:** autism spectrum disorders, propionic acid, neurotransmitters, cytokines, apoptosis, oxidative stress, gut microbiota

## Abstract

Autism spectrum disorders (ASDs) are neurodevelopmental disorders that clinically presented as impaired social interaction, repetitive behaviors, and weakened communication. The use of bee pollen as a supplement rich in amino acids amino acids, vitamins, lipids, and countless bioactive substances may lead to the relief of oxidative stress, neuroinflammation, glutamate excitotoxicity, and impaired neurochemistry as etiological mechanisms autism. Thirty young male Western albino rats were randomly divided as: Group I-control; Group II, in which autism was induced by the oral administration of 250 mg propionic acid/kg body weight/day for three days followed by orally administered saline until the end of experiment and Group III, the bee pollen-treated group, in which the rats were treated with 250 mg/kg body weight of bee pollen for four weeks before autism was induced as described for Group II. Markers related to oxidative stress, apoptosis, inflammation, glutamate excitotoxicity, and neurochemistry were measured in the brain tissue. Our results indicated that while glutathione serotonin, dopamine, gamma-aminobutyric acid (GABA), GABA/Glutamate ratio, and vitamin C were significantly reduced in propionic acid-treated group (*p* < 0.05), glutamate, IFN-γ, IL-1A, IL-6, caspase-3, and lipid peroxide levels were significantly elevated (*p* < 0.05). Bee pollen supplementation demonstrates protective potency presented as amelioration of most of the measured variables with significance range between (*p* < 0.05)–(*p* < 0.001).

## 1. Introduction

Autism spectrum disorder (ASD) is neurodevelopmental disorders that present clinically as weakened communication ability and social interaction and repetitive behaviors. These disabilities are described as a spectrum because of the significant difference in how individuals clinically present autistic features. Majority of people with ASD fall between two extremes; individuals with minimal verbal and intellectual talent, and those with average to higher intellectual ability [1]. Regardless of their place on this spectrum, each individual has an excellent chance to achieve life success following early detection and intervention of the disorder [2,3].

Although there is no cure for ASD, early diagnosis increases the awareness of ASD-associated pathologic behaviors before they progress and become severe. The amendment of core autistic features as early as possible can minimize the progression of symptoms and maximize the effects of early intervention [4].

Gut microbiota has a central role in the etiological mechanism of autism. Elevated levels of short-chain fatty acids and toxin-generating pathogenic bacteria such as *Clostridioides* species are often detected in individuals with ASD [5,6,7]. Propionic acid (PPA) is one of the most abundant short-chain fatty acids produced by *Clostridioides* species after fermentation of indigestible carbohydrates. It has many useful functions, including neuroendocrine regulation, tumor suppression, and anti-inflammation. Also, PPA represents the main mediator between the gut microbiota and the brain through the bidirectional pathway of the gut microbiota-brain axis [8,9]. However, excessive PPA can induce neurotoxic effects, such as mitochondrial oxidative stress, metabolic, and immune dysfunctions, which are all related to autism [8,10]. Choi et al. [11] demonstrate that PPA administration induced autism-like neuro-behaviors such as increased aggressive behavior, reduced probing activity, and impaired social interaction through the induction of abnormal neural cell organization, which support the validity of PPA-induced rodent model of autism [8]. Noticeably, records supports a significant role of PPA in modifying human fetal-derived neural stem cells (NSCs) modeling leading to gliosis, impaired neuro-circuitry, and inflammatory response as seen in individuals with ASD [11].

Moreover, PPA readily crosses the gut-brain barrier and affects functional brain networks, aggravating the changes in neurotransmission, neuroinflammation, and energy metabolism within the brain [12,13,14]. Recently, it was shown that an orally administered neurotoxic dose of PPA in rodents affected social skills and cognitive flexibility, and induced other alterations similar to those observed in autistic individuals [13,14,15].

Natural products, such as bee products, are often recommended by users of complementary and alternative medicine. Among the various products, bee pollen is a natural honeybee product that represents a well-known source of energy and nutrition. The health-promoting potency of bee pollen is anticipated to be due to the presence of large numbers of secondary plant metabolites, such as tocopherol, niacin, thiamine, biotin, folic acid, polyphenols, carotenoid pigments, and phytosterols [16,17,18]. The main difficulty in the application of bee pollen in phytomedicine is related to the wide interspecific variation in its composition.

The chemistry of bee pollen is greatly affected by the vegetal and geographic sources, and environmental factors, such as temperature, water, and light intensity. Different bee pollen grains have almost same nutritional value originates from their content of essential constituents [19]. Bee pollen is a rich source of the amino acids such as aspartic acid, glutamic acid, proline, leucine, lysine, and arginine [19,20]. Carbohydrates, mostly polysaccharides, including starch, pectin, sporopollenin, and cellulose, together with monosaccharides and disaccharides, constitute a considerable percentage of bee pollen [21,22]. Pollen grains contain high levels of minerals; N is the most abundant element, and K, Mg, P, Ca, S, B, Zn, Cu, Mn, and Fe are also present [19]. Pollen contains up to 10% lipids; up to 60% of the fatty acids are unsaturated acids (e.g., oleic, linoleic, and linolenic acids), and palmitic acid is the most common saturated fatty acid [23]. In recent years, pollen has been investigated by many scientists who have considered it as a natural source of healthy food, energy, and functional components for human consumption [24,25]. Pollen was proven to have therapeutic properties, with strong antioxidative, anti-inflammatory, anticarcinogenic, anti-allergenic, anti-radiation, and antitoxic activities [26,27,28]. It was even suggested that pollen has a positive effect on the treatment of neuroinflammation, and may, therefore, be used as a treatment for ASD [29].

In a recent study by Al-Yousef et al. [30], the chemical composition of bee pollen products from Wadi Al-Nahil Company, Saudi Arabia, was analyzed. The major compounds found were ethyl ester of hexadecanoic acid, eicosatrienoic acid, and 1,4-dimethyl-benzene; hexadecanoic acid (palmitic acid) and nonacosane were also found. This study also confirmed the presence of a large number of polyphenols, mainly flavonoids, and proanthocyanidins, both of which have strong antioxidant potential through direct free radical scavenging and antibacterial activities. Most interestingly, they reported that Wadi Al-Nahil bee pollen had an in vitro antibacterial effect against *Clostridioides perfringens*, a pathogenic bacterium found to be 10-fold higher in patients with autism compared with healthy controls.

This information was the source of our interest in the biochemical protective effects of bee pollen against the neurotoxic effects of propionic acid (PPA) as a metabolic products of *Clostridioides* species, known to induce persistent biochemical, behavioral and pathological autistic features in rodent model of autism [31,32]. Variables related to oxidative stress, neurotransmission, neuroinflammation, immune dysfunction, and gut microbiota were selected to evaluate the persistent biochemical and neurotoxic effects of PPA that may be ameliorated by the use of bee pollen as a protective intervention strategy.

## 2. Results

The data from the present study are presented in five tables and two figures. The descriptive changes in the selected measured variables in the three treatment groups are shown in Table 1 and Figure 1.

### 2.1. Neurotransmitters, GABA, Glutamine, Glutamate, and Caspase-3

A decrease in 5-HT, dopamine, GABA, and the GABA/Glutamate ratio was notable in Group II, with a notable restoration in Group III. In contrast, glutamate was significantly increased in Group II but was present at normal levels in Group III.

### 2.2. Neuroinflammation and Oxidative Stress Related Markers

With regard to the neuroinflammation-related parameters, it was clearly seen that although levels of the inflammatory cytokines IFN-γ, IL-1A, and IL-6, as markers of PPA neurotoxicity, were significantly elevated in Group II, the levels were much lower in Group III. No effect of bee pollen was observed on IL-12, TNF-α, and VEGF (Table 1 & Figure 1). Caspase-3, a pro-apoptotic marker, was significantly higher in Group II and lower in Group III, although still significantly higher than in Group I.

The antioxidant effect of bee pollen was clearly observed through the remarkable increase in vitamin C as antioxidant vitamin and decrease in lipid peroxides, a marker of oxidative stress.

The positive and negative correlations between the measured parameters are presented in Table 2. All measured markers were either positively or negatively correlated. The recorded positive correlations demonstrated the contribution of oxidative stress, neuroinflammation, and impaired neurotransmission, as integrated etiological mechanisms of the neurotoxicity of PPA in the rodent model of autism. The heat map provides a visual representation of the correlation; red colors indicate a more positive correlation, and blue colors indicate a more negative correlation. Significance is denoted by * (*p* < 0.05), ** (*p* < 0.01), and *** (*p* < 0.001).

The stepwise multiple regression using selected markers of apoptosis (caspase-3), glutamate excitotoxicity (Glutamate and GABA), neuroinflammation, and oxidative stress as dependent markers are shown in Table 3. The obtained data confirmed the contribution of the different predictors to the different affected signaling pathways.

The ROC analysis of the measured parameters is presented in Table 4. With a few exceptions, most of the measured markers recorded high AUC, together with high specificity and sensitivity, either as markers of the neurotoxicity of PPA or the protective effects of bee pollen. Spearman correlation matrix was calculated using GraphPad Prism version 6. Regression-relationships plots were produced using IBM SPSS version.

Figure 2 demonstrates average bacterial counts in fecal samples of control, PPA-induced rodent model of autism, and bee pollen-protected autism model. Altered gut microbiota can be easily observed as remarkable increase or decrease of certain bacterial abundance.

## 3. Discussion

This study describes an application of Spearman’s correlations in the field of toxicology to identify various associations between five neurotransmitters as measure of brain connectivity, seven cytokines as measure of neuroinflammation, one pro-apoptotic marker, and three oxidative stress-related variables in PPA-treated rats and PPA-treated rats that received a 4-week pre-treatment with bee pollen. Spearman’s correlations were displayed by using correlation heat maps, with positive and negative correlations easily identified as blue and red squares, respectively, at the intersection of two biomarkers.

In an attempt to understand the role of glutamate excitotoxicity, neuroinflammation, and oxidative stress, as three possible mechanisms related to autism induced through PPA neurotoxicity, Spearman’s correlations were tested between GABA/glutamate ratio as a marker of excitotoxicity and all the measured parameters. The significant negative correlation between GABA/glutamate ratio and lipid peroxides, an oxidative stress marker, and a positive correlation with vitamin C, an antioxidant marker, demonstrated the involvement of both compounds in the etiopathology of PPA neurotoxicity (Table 2). This was further supported through Spearman’s correlations between the absolute value of GABA as the major inhibitory neurotransmitter. In contrast with glutamate, GABA showed a significant negative correlation with Glut, the Glut/Gln ratio as a component of the glutamate/GABA/Gln cycle and lipid peroxides as a measure of oxidative stress. Moreover, it showed a positive correlation with the impaired GABA/Glut ratio as measure of imbalanced inhibitory/excitatory neurotransmission and vitamin C. This was supported by the previous of Olloquequi et al. [33] in which they prove that during excitotoxicity, the much higher release of glutamate, together with the concomitant excessive activation of glutamate receptors, can lead to the dysregulation of Ca^2+^ homeostasis, triggering the production of free radicals and oxidative stress, mitochondrial dysfunction, and eventually cell death.

Given that excitotoxicity is considered a hallmark of autism, drugs moderating different aspects of excitotoxic mechanisms may offer benefits in the treatment of individuals with autism. The obtained data have ascertained the role of excitotoxicity and oxidative stress in the neurotoxic effect of PPA and indicate the promising strategy of using bee pollen as a natural product to target the excitotoxic insult. The protective potency of bee pollen can be observed through the suppression of the elevated levels of glutamate and lipid peroxides, and the stimulation of GABA, vitamin C, and GSH (Table 1).

The unexpected negative Spearman’s correlations between glutamate and TNF-α reported in the present study can be easily related to the disruption of BBB in PPA-treated rats. It is known that permeability of the BBB is increased in response to PPA treatment and is correlated with elevated serum TNF-α levels due to the efflux of increased TNF-α from the brain to the blood [34]. This explanation is supported by the study of Mirza and Sharma [35] in which they proved BBB disruption was among the neurotoxic effects of PPA. Dysfunction of the BBB may allow PPA to bypass the barriers and enter the brain. This suggestion may also be related to the neuroprotective effect of TNF-α against glutamate excitotoxic damage in primary cortical neurons via sustained activation of nuclear factor-kappa B (NF-kappa B). The transcription factor NF-kappa B can regulate the expression of small conductance calcium-activated potassium (K (Ca)) channels. These channels reduce neuronal excitability and, as such, may confer neuroprotective effects against neuronal overstimulation. The suggested efflux of TNF-α through the disrupted BBB leads to lower levels of brain TNF-α and decreased (K (Ca)) channel expression, which render neurons more susceptible to glutamate-mediated excitotoxic cell death [36]. This may support the obtained Pearson’s negative correlation.

When in the brain, PPA may modify neural circuitry through the induction of the excitatory glutamatergic transmission and the impedance of GABAergic transmission, which may lead to imbalanced E/I neural circuitry, a repeatedly reported feature of autism [37,38]. Inhibiting GABAergic transmission as an important neurotransmitter for pre-pulse inhibition (PPI) can result in a decrease in PPI upon PPA administration. Also, PPA can lead to intracellular acidification by uncoupling gap junctions between neurons and disrupting synaptic transmission in the prefrontal cortex and sensorimotor cortex [39]. Collectively, these effects of PPA on neural circuitry can explain the deficits in sensorimotor activity and the altered cognitive functioning that manifest as autistic features in the PPA-induced rodent model of autism. Again, this explanation is supported through the positive and negative correlations presented between GABA/Glut ratio and TNF-α, vitamin C, and lipid peroxides (Table 2).

Several associations reported between IFN-γ, as a neuroinflammatory marker, and pro-apoptotic, and neurotransmission/oxidative stress markers were expected based on known physiological processes. The data presented in Table 2 demonstrate that although IFN-γ was negatively correlated with NE, DA, 5-HT, GABA, GABA/Glut, and vitamin C, it was positively correlated with glutamate, Glut/Gln, caspase-3, and VEGF P. Neuroinflammation, a typical neurotoxic effect, can trigger oxidative stress through several mechanisms, such as the production of high levels of ROS by activated astrocytes and microglia and through the activation of cyclooxygenase (COX) and lipoxygenase (LOX) pathways [40,41,42]. The overproduction of ROS activates microglia and promotes further cytokine production [43], leading to a positive correlation between IFN-γ and lipid peroxides. This confirmed both the implication that IFN-γ was involved in stimulating the release of ROS from microglia in PPA-treated rats and the efficacy of bee pollen as an antioxidant and anti-inflammatory product, through the remarkable correction of both signaling related variables [41]. This is supported by the recent work of Erten [44] in which attenuation of postnatal PPA-induced oxidative stress and neuroinflammation in rats was treatable and accompanied an improvement in the autism-like behavioral phenotypes induced.

As reduced levels of neurotransmitters, such as NE, 5-HT, DA, and GABA, are often associated with the pathogenesis of autism. The negative Spearman’s correlations of these four variables and IFN-γ, as a marker of neuroinflammation, can indicate the contribution of neuroinflammation in the anxiety-like behaviors related to these neurotransmitters in the phenotypes of autism [45,46].

Oxidative stress, glutamate excitotoxicity, and neuroinflammation are among the basic events in many neurological disorders. These three etiological mechanisms are related to epilepsy, as co-morbidities of autism that lead to cell death. Caspase-3, a pro-apoptotic marker, was found to be positively correlated with Glut and the Glut/Gln ratio, and negatively correlated with GABA and the GABA/Glut ratio, all of which are glutamate-mediated excitotoxicity markers [47]. The positive Spearman’s correlation between caspase-3 and lipid peroxides, together with the negative correlation between caspase-3 and vitamin C, an antioxidant vitamin, confirmed the association of apoptosis with oxidative stress. This further supported the contribution of this pathogenic triad to the neurotoxicity of PPA and its relationship to neuronal cell death, indicated by the elevation of caspase-3, a marker of apoptosis, a fourth etiological mechanism in autism. The unexpected negative Spearman’s correlations reported in the present study between caspase-3, IL-12, and TNF-α explain the disruption of BBB in PPA-treated rats. Serum studies from ASD patients provide additional proof linking PPA to the disorder, as ASD patients have metabolic dysfunction including oxidative stress, imbalanced GABA/glutamate ratio, neuroinflammation, and mitochondrial dysfunction. Elevated levels of PPA may be a gut-derived factor involved in ASD patients [11,15].

In an attempt to explain the protective mechanism of bee pollen, it should be considered that bee pollen is rich with multiple antioxidant vitamins, such as vitamins C, E, and β-carotene, as pro-vitamin A, together with vitamin B complex and vitamin D [48,49]. Among these vitamins, vitamin E appears to be better preserved during storage compared with vitamin C and β-carotene. The obtained ameliorative effects of bee pollen presented in Figure 1 may be related to the abundance of these antioxidant vitamins in bee pollen. Among the measured variables, vitamin C was significantly elevated in Group III (bee pollen-treated autism model group) compared with Group II (autism model group). The effectiveness of vitamin C as a component of bee pollen that protects against PPA neurotoxicity was clearly observed through the use of Spearman’s correlation as statistical tool (Table 1): although vitamin C was negatively correlated with IFN-γ, caspase-3, lipid peroxides, and glutamate, as markers of pro-inflammation, apoptosis, oxidative stress, and excitotoxicity, respectively, which are neurotoxic effects of PPA, it was positively correlated with GABA, the GABA/Glut ratio, noradrenergic and dopaminergic neurotransmission, and GSH as markers of balanced E/I neurotransmission, normal emotional reactivity, and antioxidant capacity [50]. This is heavily supported by a recent study by Ambrogini et al. [47] in which it was proved that vitamin E, as a major component of bee pollen, was effective in counteracting oxidative stress, glutamate excitotoxicity, neuroinflammation, and apoptosis, four of the molecular events related to autism. Besides its antioxidant effects, vitamin E can modulate enzymes, signal transduction pathways, and transcription factors associated with the inflammatory response, such as the NF-kappa B pathway of activated mononuclear cells, and the LOX-dependent and COX-dependent pathways of arachidonic acid oxidation to generate inflammatory and vasoactive eicosanoids, which are lipid mediators related to the neurotoxicity of PPA and pathology of autism [51].

As Spearman’s correlation is a measure of the strength of association between different variables, multiple regression analysis is used to acquire more quantitative assessments about how the change in one variable (dependent; or responders) relies on the alteration of other variables (independent; or predictors).

It can be easily noticed that the more independents (or predictors) that are variables related to the different etiological mechanism the higher the significance (*p*-value) and R^2^ value for a dependent variable. For example, although 79.1% of the change in caspase-3 was affected by the significant elevation of glutamate as marker of excitotoxicity, 99.9% of the change in this apoptotic marker was affected by the panel of independent variables, representing impaired neurochemistry (Glut, Gln, DA) and neuroinflammation (IL-12, IL-10, IFN-γ). The same trend was followed by all the test variables of the current study, as shown in Table 3.

The ROC analysis for all the measured variables is shown in Table 4. With few exceptions, all the measured variables showed high AUC, together with satisfactory specificity and sensitivity as markers of PPA neurotoxicity or the therapeutic potency of bee pollen.

Collectively, the protective effects of bee pollen reported in the current study can find support in a recent study Li et al. [52]. They demonstrated that bee pollen protective effects against intestinal barrier impairment by strengthening epithelial integrity and tight junction losses induced by toxic insults through the up-regulation of anti-oxidants and down-regulation of inflammatory cytokines (TNF-α and IL-6) [52]. This might help to suggest that the protective effects of bee pollen as prebiotic takes place through the gut microbiome-brain axis as contributing factor in the etiology of autistic features in rodent model.

In Figure 2, it is clearly noticed that PPA induced a remarkable alteration at the genus level of gut flora of the treated animals. Increase of unclassified *Staphylococcus* and/or *Bacilli, Enterobacteriaceae* genus in the n PPA- rodent model of ASD (Group II) is in good agreement with multiple clinical and experimental findings demonstrating the elevation of the three genus in ASD patients and rodent model [53,54,55]. Additionally, the remarkable increase in *Bactericides sp. streptococci* and *enterococci* in PPA-treated rats is in good agreement with the work of Lee et al. [54] in which they reported almost 3 folds higher of these bacteria in ASD patients. The remarkable ameliorating effects of the bee pollen-protected group could be supported by considering the antimicrobial activity of bee pollen extracts on *Staphylococcus aureus*, *Escherichia coli* and other bacterial strains [56,57]. This might help to support the increasing interest regarding the effectiveness of the antimicrobial properties of bee pollen, due to the emerging antibiotic resistance developed by different pathogens as a future expected global health hazard.

## 4. Materials and Methods

Bee pollen was purchased from a branch of Wadi Al-Nahil, in Riyadh, Saudi Arabia in June 2017, under the trade name “bee pollen, 100% natural bee pollen first elite”. It was imported by Wadi Al-Nahil, one of the largest marketing companies in Saudi Arabia (www.wadialnahil.net, accessed on 18 June 2022).

### 4.1. Animals

Thirty young Western albino male rats weighing 100 ± 20 g; at the age of 4 weeks were obtained from the Experimental and Surgery Animal Laboratory, King Khalid Hospital, King Saud University. Rats were housed individually in stainless steel cages under controlled environmental conditions of 25 °C, 12 h day/night cycle, the humidity of 50% ± 5.

### 4.2. Experimental Design

A pre-determination of sample size calculation was not performed. The rats were randomly and blindly divided into three groups: Group I, the control group, comprising control healthy rats; Group II, the autism rodent model group, in which rats were orally administered phosphate-buffered saline (PBS) for 25 days then received 250 mg PPA/kg body weight/day for three days from experimental days 26 to 28; Group III, the bee pollen-protected group, in which the rats were treated with orally administered 250 mg/kg body weight of bee pollen from experimental days 1 to 25 before autism was induced as described for Group II (i.e., oral administration of PPA from experimental day 26–28). The duration used was selected up to numerous previous studies showing average duration between three weeks-one month of bee pollen treatment [29,58].

### 4.3. Ethics Approval

The protocol of this work was approved by and carried out in accordance to the guidelines of Graduate Studies and Scientific Research Ethical Committee of Bioethics in King Saud University Ref Number: 4/67/352670. Our study was carried out in compliance with the ARRIVE guidelines.

### 4.4. Collection of Brain Samples

The rats were anesthetized with carbon dioxide and decapitated. The brain was removed, washed with distilled water. The whole brain was homogenized in 10 *w*/*v* bidistelled water using electric homogenizer.

### 4.5. Quantification of Neurotransmitters (Norepinephrine, Dopamine, and Serotonin)

The concentrations of norepinephrine (NE) and dopamine (DA) were determined in brain homogenates using a Competitive ELISA kit from MyBioSource. The concentration of serotonin (5-HT) was determined using the Quantitative Sandwich ELISA kit from MyBioSource.

### 4.6. Quantification of GABA, Glutamine, Glutamate, and Caspase-3

GABA was quantitatively determined by using an ELISA kit from ALPCO Diagnostics (Salem, NH, USA). Rat brain glutamine (Gln), glutamate (Glut), and caspase-3 were measured independently by using ELISA kits from Cusabio (Wuhan, China).

### 4.7. Quantification of Cytokines

The MILLIPLEX^®^ MAP (Merck KGaA, Darmstadt, Germany) kit for Mouse Cytokine/Chemokine, Magnetic Bead panel, was used following the manufacturer’s instructions for the quantification of the cytokines in the brain homogenate of rats. The panel measured the following cytokines: interferon-γ (IFN-γ), interleukin-1 alpha (IL-1α), IL-6, IL-10, IL-12 p70, vascular endothelial growth factor (VEGF), and tumor necrosis factor α (TNF-α).

### 4.8. Quantification of Oxidative Stress Markers

Glutathione was assayed by the method of Beutler et al. [59] using 5,5′-dithiobis 2-nitrobenzoic acid (DTNB) with sulfhydryl compounds to produce a relatively stable yellow color. Lipid oxidation was estimated by the formation of thiobarbituric acid reactive substances (TBARS) by the method of Ruiz-Larrea et al. [60] Vitamin C level was estimated using the method described by Jagota and Dani [61].

### 4.9. Bacterial Counts in Fecal Samples

Animal fecal samples were collected and diluted in phosphate-buffered saline (1:10 *w*/*v*) [62]. The mixture was vortexed for 1 min, and then centrifuged at 3000× *g* rpm for 3 min at 4 °C 100 μL of each sample was spotted on 5 different selective media plates which include: Nutrient agar, Mac Conkey, Blood agar, Cycloserine- Cefoxitine Fructose Agar(CCFA), and Bile Esculin Agar (BBE) [62,63]. Plates were incubated at 37 °C for 18–24 h aerobically except CCFA and BBE which were incubated in anaerobic conditions (5% CO_2_). Colonies on each media plates were studied macroscopically, numbered, and then microscopically identified by gram staining technique.

### 4.10. Statistical Analyses

All the measured markers were analyzed by using Spearman correlations and stepwise multiple regressions as statistical tools to help elucidate the relationship between brain connectivity, glutamate excitotoxicity, oxidative stress, and neuroinflammation as neurotoxic signals related to PPA and the use of bee pollen as a protective agent. This can help to elucidate the etiopathological mechanisms of ASD and the success of the protective approach of bee pollen as natural product safe supplement.

The results in the present study were expressed as the mean ± S.D. All statistical comparisons between the three groups were performed by using one-way analysis of variance (ANOVA) tests with Dunnett’s test for multiple comparisons. The statistical analyses were computed by using Statistical Package for the Social Sciences (SPSS, Chicago, IL, USA), and significance was assigned for *p* values of <0.05. Spearman’s correlation analysis was used to measure the statistical dependence between two variables because the majority of biomarker concentrations were not normally distributed. Spearman’s correlation coefficients (r) and *p*-values were determined between all the biochemical variable markers. Multiple regression stepwise analysis was performed between the measured parameters. Receiver operating characteristics (ROC) curve analysis was also performed. The area under the curve (AUC), the degrees of sensitivity and specificity, and cutoff values were calculated.

## 5. Conclusions

This study has ascertained the integration and interaction between apoptosis, inflammation, glutamate toxicity, oxidative stress, and impaired neurochemistry in either the neurotoxic effects of PPA or the protective effects of bee pollen. As all the mechanisms mentioned above are prominently recorded as pathophysiological phenomena in ASD, bee pollen can be suggested as a protective strategy, which should be confirmed by future studies.

## Figures and Tables

**Figure 1 metabolites-12-00571-f001:**
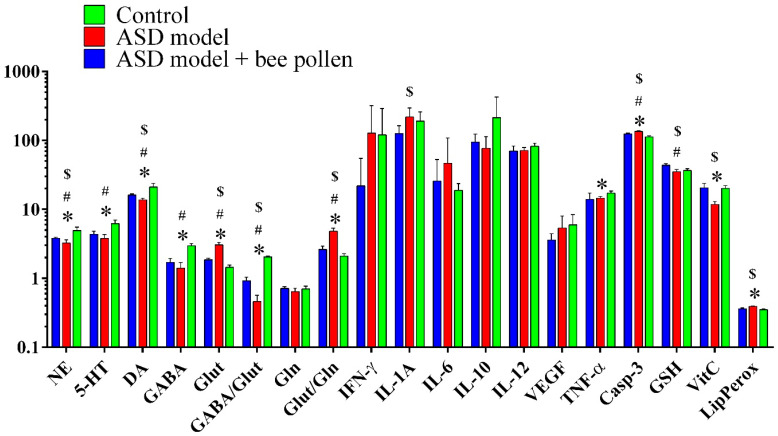
Levels of the biomarkers used in ASD (Group II; APP-treated), ASD/bee pollen-protected (Group III), and control (group I) rats. * denotes a statistically significant difference between groups I and II, # denotes a statistically significant difference between groups I and III, and $ denotes statistically significant difference between groups II and III. NE: norepinephrine, 5-HT: serotonin, DA: dopamine, GABA: gamma-aminobutyric acid, Glut: glutamate, Gln: glutamine, IFN-γ: interferon gamma, IL-x: interleukin x, VEGF: vascular endothelial growth factor, TNF-α: tumor necrosis factor alpha, Casp-3: caspase 3, GSH: glutathione, VitC: vitamin C, LipPerox: lipid peroxidase.

**Figure 2 metabolites-12-00571-f002:**
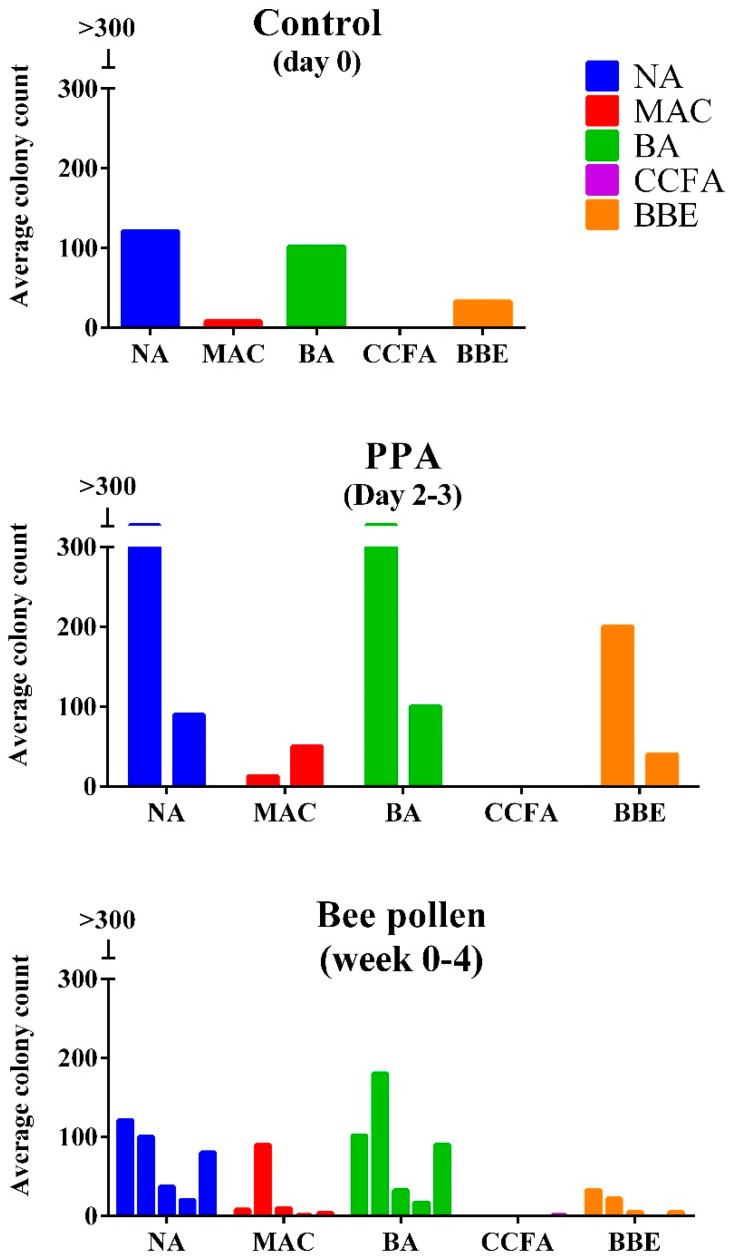
Average Bacterial counts in fecal samples of control, PPA-induced rodent model of autism, and bee pollen-protected autism model. NA: nutrient agar, MAC: MacConkey’s agar, BA: blood agar, CCFA: cycloserine cefoxitin fructose agar, BBE: *Bacteroides* bile esculin agar. Nutrient agar—Gram-positive/Gram-negative rods and cocci; MacConkey—*Enterobacteriaceae* (Gram-negative rods, lactose fermenters); Blood agar—Gram-positive/Gram-negative rods and cocci; CCFA (Cycloserine Cefoxitin Fructose Agar)—*Clostridioides difficile;* BBE (Bile Esculin Agar)*—Bacteroidetes*, streptococci, and enterococci.

**Table 1 metabolites-12-00571-t001:** Mean ± S.D. of the measured variables in PPA-treated (Group II) and bee pollen-protected animals (Group III) compared to controls. GABA: gamma-aminobutyric acid, IFN-γ: interferon gamma, IL-x: interleukin x, VEGF: vascular endothelial growth factor, TNF-α: tumor necrosis factor alpha. * denotes significance between ASD model or BP-protected ASD model and control; ^#^ denotes significance between BP-protected ASD model and ASD model; ^$^ denotes significance between the three studied groups.

Variables	Control	ASD Model	Bee Pollen-Protected ASD Model
Norepinephrine (ng/100 mg)	4.92 ± 0.618	3.25 ± 0.36 *^,#^	3.79 ± 0.15 *^,#^
Serotonin (ng/100 mg)	6.20 ± 0.78	3.78 ± 0.56 *^,#^	4.36 ± 0.42 *^,#^
Dopamine (ng/100 mg)	21.12 ± 2.72	13.69 ± 0.80 *^,#^	16.09 ± 0.71 *^,#^
GABA (ng/100 mg)	2.95 ± 0.24	1.40 ± 0.29 *^,#^	1.70 ± 0.24 *^,#^
Glutamate (µg/mg)	1.44 ± 0.11	3.05 ± 0.27 *^,#^	1.85 ± 0.10 *^,#^
GABA/Glutamine	2.04 ± 0.06	0.46 ± 0.11 *^,#^	0.92 ± 0.12 *^,#^
Glutamine (µg/mg)	0.70 ± 0.07	0.64 ± 0.08 *^,#^	0.71 ± 0.05 *^,#^
Glutamate/Glutamine	2.09 ± 0.19	4.80 ± 0.52 *^,#^	2.62 ± 0.30 *^,#^
IFN-γ (pg/mL)	120.71 ± 167.85	127.09 ± 192.57 ^$^	21.86 ± 32.80 ^$^
IL-1A (pg/mL)	190.54 ± 68.50	217.85 ± 76.89 *^,#^	125.94 ± 37.27 *^,#^
IL-6 (pg/mL)	18.87 ± 4.69	46.44 ± 61.93 ^$^	25.64 ± 27.15 ^$^
IL-10 (pg/mL)	213.86 ± 212.98	76.76 ± 36.44 ^$^	94.08 ± 28.87 ^$^
IL-12 (pg/mL)	82.07 ± 8.65	71.03 ± 7.57 ^$^	70.11 ± 13.10 ^$^
VEGF (pg/mL)	5.95 ± 2.43	5.36 ± 2.64 ^$,#^	3.58 ± 0.85 *^,#^
TNF-α (pg/mL)	17.11 ± 1.39	14.50 ± 0.91 ^$,#^	13.95 ± 3.27 *^,#^
Caspase-3 (u/100 mg)	112.66 ± 4.20	134.79 ± 3.27 *^,#^	124.23 ± 2.89 *^,#^
Glutathione (µg/mL)	36.44 ± 2.43	35.07 ± 2.83 *^,$^	43.65 ± 2.09 *^,#^
Vitamin C (µg/mL)	20.16 ± 2.00	11.74 ± 1.16 *^,#^	20.42 ± 3.42 ^$,#^
Lipid peroxides	0.35 ± 0.01	0.39 ± 0.01 *^,#^	0.36 ± 0.01 ^$,#^

**Table 2 metabolites-12-00571-t002:** Spearman correlation matrix. Only significant correlations are shown (*p* value < 0.05). Correlation heatmap depicts perfect positive (darks blue) and negative (dark red) correlations and no correlation (white). Significance heatmap depicts *p* values ranging from 0.000 (green) to 0.05 (orange). NE: norepinephrine, 5-HT: serotonin, DA: dopamine, GABA: gamma-aminobutyric acid, Glut: glutamate, Gln: glutamine, IL-12: interleukin 12, VEGF: vascular endothelial growth factor, TNF-α: tumor necrosis factor alpha, GSH: glutathione, VitC: vitamin C, MDA: 3,4-methylenedioxyamphetamine.

Variable	Correlated Variables	Spearman Correlation (R)	*p* Value	Variable	Correlated Variables	Spearman Correlation (R)	*p* Value
**Interferon** **ϒ**	NE	−0.829	0.000	**Norepinephrine**	5-HT	0.82	0.000
	5-HT	−0.69	0.002		DA	0.888	0.000
	DA	−0.818	0.000		Caspase 3	−0.865	0.000
	Caspase 3	0.771	0.000		GABA	0.919	0.000
	GABA	−0.686	0.002		Glut	−0.91	0.000
	Glut	0.756	0.000		GABA/Glut	0.95	0.000
	GABA/Glut	−0.748	0.000		Glut/Gln	−0.85	0.000
	Glut/Gln	0.735	0.001		IL-12	0.473	0.047
	VEGF	0.652	0.006		TNF-α	0.557	0.016
	TNF-α	0.649	0.007		VitC	0.634	0.005
	VitC	−0.646	0.004		MDA	−0.702	0.001
	MDA	0.717	0.001				
**Caspase 3**	GABA	−0.846	0.000	**Dopamine**	Caspase 3	−0.913	0.000
	Glut	0.856	0.000		GABA	0.839	0.000
	GABA/Glut	−0.874	0.000		Glut	−0.851	0.000
	Glut/Gln	0.872	0.000		GABA/Glut	0.89	0.000
	IL-12	−0.527	0.024		Glut/Gln	−0.86	0.000
	TNF-α	−0.499	0.035		TNF-α	0.508	0.031
	VitC	−0.633	0.005		VitC	0.596	0.009
	MDA	0.71	0.001		MDA	−0.778	0.000
**Serotonin**	DA	0.827	0.000	**GABA**	Glut	−0.781	0.000
	Caspase 3	−0.839	0.000		GABA/Glut	0.926	0.000
	GABA	0.859	0.000		Glut/Gln	−0.778	0.000
	GABA/Glut	0.853	0.000		VitC	0.521	0.027
	Glut/Gln	−0.824	0.000		MDA	−0.605	0.008
	TNF-α	0.547	0.019	**IL-1A**	GSH	−0.75	0.001
	VitC	−0.658	0.003		MDA	0.528	0.035
**GABA/Glut**	Glut/Gln	−0.862	0.000		TNF-α	0.746	0.000
	TNF-α	0.486	0.041	**GSH**	VitC	0.552	0.018
	VitC	0.622	0.006				
	Lipid peroxides	−0.775	0.000	**VitC**	MDA	−0.553	0.017

**Table 3 metabolites-12-00571-t003:** Regression analysis showing the relationships between five dependent variables and all other biomarkers. NE: norepinephrine, 5-HT: serotonin, DA: dopamine, GABA: gamma-aminobutyric acid, Glut: glutamate, Gln: glutamine, IFN-γ: interferon gamma, IFN-γ: interferon upsilon, IL-x: interleukin x, VEGF: vascular endothelial growth factor, TNF-α: tumor necrosis factor alpha, GSH: glutathione.

Dependent Variable	Predictor Variable	Coefficient	S.E.	*p* Value	Adjusted R^2^	95% CI		Dependent Variable	Predictor Variable	Coefficient	S.E.	*p* Value	Adjusted R^2^	95% CI	
						Lower	Upper							Lower	Upper
**Caspase 3**	Glut	10.044	1.612	0.000	0.791	6.398	13.690	**VEGF**	IL-10	0.011	0.001	0.000	0.853	0.008	0.014
	Glut	8.641	1.397	0.000	0.868	5.419	11.862		IL-10	0.011	0.001	0.214	0.912	0.009	0.014
	IL-12	−0.309	0.123	0.000		−0.594	−0.025		IL-1A	0.012	0.005	0.000		0.002	0.023
	Glut	6.986	0.712	0.000	0.972	5.303	8.668		IL-10	0.011	0.001	0.000	0.946	0.009	0.013
	IL-12	−0.408	0.060	0.000		−0.549	−0.266		IL-1A	0.012	0.004	0.014		0.003	0.020
	Gln	64.781	11.733	0.001		37.038	92.524		IL-6	−0.019	0.008	0.045		−0.038	0.000
	Glut	3.695	1.219	0.023	0.987	0.712	6.678	**NE**	GABA	1.205	0.158	0.000	0.851	0.848	1.563
	IL-12	−0.441	0.043	0.000		−0.545	−0.336		GABA	0.679	0.251	0.557	0.904	0.099	1.259
	Gln	63.796	8.105	0.000		43.965	83.627		DA	0.138	0.057	0.027		0.007	0.269
	IFN-ϒ	0.143	0.049	0.026		0.024	0.262	**IFN-γ**	Glut	22.035	2.923	0.000	0.848	15.423	28.646
	Glut	2.785	0.897	0.027	0.994	0.479	5.092	**IL-1A**	IL-12	2.832	1.228	0.047	0.301	0.053	5.611
	IL-12	−0.470	0.031	0.000		−0.550	−0.390		IL-12	4.173	0.967	0.028	0.650	1.943	6.404
	Gln	67.868	5.745	0.000		53.101	82.635		Glut/Gln	24.766	7.840	0.003		6.688	42.844
	IFN-ϒ	0.188	0.037	0.004		0.093	0.283	**IL-10**	VEGF	80.798	10.535	0.000	0.853	56.965	104.631
	IL-10	0.004	0.002	0.039		0.000	0.008		VEGF	83.026	8.106	0.353	0.914	64.333	101.719
	Glut	2.797	0.356	0.001	0.999	1.809	3.784		IL-1A	−1.052	0.388	0.000		−1.946	−0.158
	IL-12	−0.516	0.015	0.000		−0.558	−0.474	**GSH**	TNF-α	−1.134	0.480	0.042	0.314	−2.220	−0.048
	Gln	74.889	2.637	0.000		67.568	82.211		TNF-α	−1.288	0.367	0.000	0.609	−2.133	−0.443
	IFN-ϒ	0.221	0.016	0.000		0.176	0.265		IFN-ϒ	−0.130	0.047	0.008		−0.238	−0.023
	IL-10	0.005	0.001	0.001		0.003	0.007	**Vitamin C**	Glut/Gln	−3.272	0.864	0.004	0.571	−5.227	−1.317
	DA	0.337	0.064	0.006		0.160	0.514		Glut/Gln	−4.571	0.898	0.000	0.714	−6.641	−2.501
**Glut**	Glut/Gln	0.706	0.040	0.000	0.969	0.616	0.796	**5-HT**	GABA/Glut	1.385	0.220	0.000	0.794	0.887	1.883
	Glut/Gln	0.685	0.014	0.000	0.996	0.653	0.717	**DA**	NE	3.149	0.436	0.000	0.836	2.162	4.136
	Gln	2.869	0.348	0.000		2.067	3.670	**GABA**	GABA/Glut	0.923	0.069	0.000	0.947	0.767	1.080
	Glut/Gln	0.713	0.009	0.000	0.999	0.692	0.735	**IL-12**	TNF-α	2.920	0.786	0.005	0.561	1.142	4.698
	Gln	2.581	0.188	0.000		2.137	3.025	**TNF-α**	IL-12	0.207	0.056	0.005	0.561	0.081	0.334
	IFN-γ	0.000	0.000	0.002		0.000	0.000	**MDA**	Glut	0.018	0.004	0.002	0.627	0.009	0.028

**Table 4 metabolites-12-00571-t004:** ROC-Curve of all variables in the PPA-treated and Bee pollen-protected groups. GABA: gamma-aminobutyric acid, Glut: glutamate, Gln: glutamine, IFN-γ: interferon gamma, IFN-γ: interferon upsilon, IL-x: interleukin x, VEGF: vascular endothelial growth factor, TNF-α: tumor necrosis factor alpha, GSH: glutathione.

Parameters	Groups	AUC	Cut-off Value	Sensitivity	Specificity	*p*-Value	95% CI
**IFN-** **ϒ (pg/100 mg)**	PPA-acute	1	103.755	100.0%	100.0%	0.004	1.000–1.000
	Pollen	0.694	82.67	100.0%	50.0%	0.262	0.369–1.020
**Noradrenaline (ng/100 mg)**	PPA-acute	1	3.955	100.0%	100.0%	0.004	1.000–1.000
	Pollen	1	4.135	100.0%	100.0%	0.004	1.000–1.000
**Serotonin (ng/100 mg)**	PPA-acute	1	4.795	100.0%	100.0%	0.004	1.000–1.000
	Pollen	1	5.105	100.0%	100.0%	0.004	1.000–1.000
**Dopamine (ng/100 mg)**	PPA-acute	1	16.52	100.0%	100.0%	0.004	1.000–1.000
	Pollen	1	17.62	100.0%	100.0%	0.004	1.000–1.000
**Caspase-3 (u/100 mg)**	PPA-acute	1	124.76	100.0%	100.0%	0.004	1.000–1.000
	Pollen	1	119.845	100.0%	100.0%	0.004	1.000–1.000
**GABA (ng/100 mg)**	PPA-acute	1	2.275	100.0%	100.0%	0.004	1.000–1.000
	Pollen	1	2.4	100.0%	100.0%	0.004	1.000–1.000
**Glut (µg/mg)**	PPA-acute	1	2.19	100.0%	100.0%	0.004	1.000–1.000
	Pollen	1	1.67	100.0%	100.0%	0.004	1.000–1.000
**GABA/Glut**	PPA-acute	1	1.29	100.0%	100.0%	0.004	1.000–1.000
	Pollen	1	1.52	100.0%	100.0%	0.004	1.000–1.000
**Gln (µg/mg)**	PPA-acute	0.722	0.655	66.7%	66.7%	0.2	0.421–1.023
	Pollen	0.569	0.665	83.3%	50.0%	0.689	0.221–0.918
**Glut/Gln**	PPA-acute	1	3.295	100.0%	100.0%	0.004	1.000–1.000
	Pollen	0.931	2.14	100.0%	66.7%	0.013	0.787–1.074
**IFN-γ (pg/mL)**	PPA-acute	0.5	25.38	50.0%	66.7%	1	0.153–0.847
	Pollen	0.833	8.445	75.0%	100.0%	0.088	0.524–1.143
**IL-1A (pg/mL)**	PPA-acute	0.633	207.35	50.0%	80.0%	0.465	0.288–0.979
	Pollen	0.88	146.235	80.0%	80.0%	0.047	0.662–1.098
**IL-6 (pg/mL)**	PPA-acute	0.521	23.195	50.0%	83.3%	0.915	0.070–0.972
	Pollen	0.6	16.88	60.0%	83.3%	0.584	0.205–0.995
**IL-10 (pg/mL)**	PPA-acute	0.812	122.27	100.0%	75.0%	0.149	0.465–1.160
	Pollen	0.875	119.215	100.0%	75.0%	0.083	0.608–1.142
**IL-12 (pg/mL)**	PPA-acute	0.833	72.835	66.7%	100.0%	0.055	0.593–1.073
	Pollen	0.861	72.13	83.3%	100.0%	0.037	0.604–1.119
**VEGF (pg/mL)**	PPA-acute	0.597	4.5	66.7%	66.7%	0.575	0.253–0.941
	Pollen	0.889	3.72	66.7%	100.0%	0.025	0.697–1.081
**TNF-α (pg/mL)**	PPA-acute	0.944	15.095	83.3%	100.0%	0.01	0.814–1.075
	Pollen	0.833	14.845	83.3%	100.0%	0.055	0.535–1.132
	Pollen	0.861	54.89	100.0%	66.7%	0.037	0.645–1.077
**GSH (µg/mL)**	PPA-acute	0.667	37.41	83.3%	50.0%	0.337	0.346–0.987
	Pollen	1	39.565	100.0%	100.0%	0.004	1.000–1.000
**Vitamin C (µg/mL)**	PPA-acute	1	15.26	100.0%	100.0%	0.004	1.000–1.000
	Pollen	0.583	20.485	66.7%	66.7%	0.631	0.234–0.933
	Pollen	0.639	0.355	66.7%	66.7%	0.423	0.301–0.977

## Data Availability

Data is contained within the article.

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
