# Peer review of "Protective Effects of Bee Pollen on Multiple Propionic Acid-Induced Biochemical Autistic Features in a Rat Model"

_metabolites, 2022, doi:10.3390/metabo12070571_

Round 1

Reviewer 1 Report

Alfawaz et al. demonstrate that bee pollen has a protective effect against propionic acid-induced autism in a rat model. The manuscript is well-written, interesting and innovative. Overall a good manuscript with several aspects of novelty. However, I have some doubts regarding the translation of an animal model to a human setting. Here are my comments.

1. It is a known fact that the administration of propionic acid in animal models, especially rat, induced behavioral changes reminiscent of autism spectrum disorders. However, which role does propionic acid play in the development of autism in humans? Are there any data available? Please discuss this important point in the introduction!

2. To which extent can your animal model simulate conditions in humans? Please discuss!

3. You used bee pollen of a single origin. For reasons of reproducibility, it would have been important to use bee pollen of different origins, at least n=3. Or do you have any data proving that bee pollens all have similar compositions, hence discarding the need to perform the experiment with several bee pollens?

Author Response

Reviewer 1

Alfawaz et al. demonstrate that bee pollen has a protective effect against propionic acid-induced autism in a rat model. The manuscript is well-written, interesting and innovative. Overall a good manuscript with several aspects of novelty. However, I have some doubts regarding the translation of an animal model to a human setting. Here are my comments.

  1. It is a known fact that the administration of propionic acid in animal models, especially rat, induced behavioral changes reminiscent of autism spectrum disorders. However, which role does propionic acid plays in the development of autism in humans? Are there any data available? Please discuss this important point in the introduction!

Done and a statement support the role of PPA in the development of autism in humans was inserted with the appropriate citation. You can find highlighted in blue within the text.(Lines 69-72) in the introduction.

  1. To which extent can your animal model simulate conditions in humans? Please discuss!

Done and a statement support the simulation of PPA rodent model to conditions in human was inserted with the appropriate citation in the discussion (Lines 316-320). You can kindly find highlighted in blue.

  1. You used bee pollen of a single origin. For reasons of reproducibility, it would have been important to use bee pollen of different origins, at least n=3. Or do you have any data proving that bee pollens all have similar compositions, hence discarding the need to perform the experiment with several bee pollens?

A statement demonstrates the more or less nutritional value and composition of bee pollen was inserted with appropriate reference. You can find highlighted in blue within the text (Lines 90- 91).

Reviewer 2 Report

Here are my comments about this submitted manuscript untitled:” Protective effects of bee pollen on multiple propionic acid-induced biochemical autistic features in a rat model” by Alfawaz et al.:

-        How did the authors choose the experimental set-up of their in vivo experiments? Did they try to treat the rats for more or less than 4 weeks with bee pollen? This should be at least explained.

-        The end of the abstract describing the results should be more detailed, included P values please.

-        Introduction part is very well written, well done to the authors, that’s indeed very clear.

-        The presentation of the results in Figure 1 should be modified to a more readable/classical histogram graph.

-        May the rest of the results also be presented in another way to increase readability? For example by showing a HeatMap ?

-        Minor typos can be found all over the manuscript and should be corrected.

Author Response

Reviewer 2

Here are my comments about this submitted manuscript untitled:” Protective effects of bee pollen on multiple propionic acid-induced biochemical autistic features in a rat model” by Alfawaz et al.:

-        How did the authors choose the experimental set-up of their in vivo experiments? Did they try to treat the rats for more or less than 4 weeks with bee pollen? This should be at least explained.

- Done and the reason for the setup of our in vivo experiments was explained and supported with two relevant references. You can find highlighted in green within the text (Lines 411-412).

-        The end of the abstract describing the results should be more detailed, included P values please.

- Done and highlighted inblue

-        Introduction part is very well written, well done to the authors, that’s indeed very clear.

- Thanks a lot for your positive response

-  The presentation of the results in Figure 1 should be modified to a more readable/classical histogram graph.

- Done. Units were removed and lipid peroxidase was abbreviated to allow readability on a horizontal axes.

-        May the rest of the results also be presented in another way to increase readability? For example by showing a HeatMap?

Response: Since Table 1 data are also shown in figure 1, no further modification was introduced to table 1. Tables 2 & 3 have not also been modified (except to correct typos and the like) since heatmaps were used in both in the original manuscript. A heatmap feature was introduced to table 4. Table 5 was converted into a figure (figure 2).  

Minor typos can be found all over the manuscript and should be corrected.

Reviewer 3 Report

Please see the attached file. Specific comments were annotated in the PDF file due to absence of line numbers.

Author Response

Reviewer 3:

Please see the attached file. Specific comments were annotated in the PDF file due to absence of line numbers.

- Thanks for your appreciated efforts in revising our manuscript. All the requested changes were done and you can find highlighted in blue within the text.

The English languages was corrected along the manuscript

Round 2

Reviewer 1 Report

The authors responded to all my requests, I have no further comments.